# Growth Characteristics of Long-Nosed Skate *Dipturus oxyrinchus* (Linnaeus, 1758) Inhabiting the Northeastern Mediterranean Sea

**DOI:** 10.3390/ani12233443

**Published:** 2022-12-06

**Authors:** Nuri Başusta, Fatih Volkan Ozel

**Affiliations:** 1Faculty of Fisheries, Firat University, 23119 Elazig, Turkey; 2Graduate School of Natural and Applied Sciences, Firat University, 23119 Elazig, Turkey

**Keywords:** age, growth, *Dipturus oxyrinchus*, long-nosed skate, Northeastern Mediterranean Sea

## Abstract

**Simple Summary:**

Skates and rays generally have low fecundity, delayed maturation age and slow growth rates. These life history traits make them very vulnerable to commercial fisheries’ activities, although they are not usually target species. Understanding age and growth is important for stock assessment of the long-nose skate. We believe that determining these can be the first step toward correct management actions. Little is known about the growth characteristics of the long-nose skate in the Northeastern Mediterranean Sea. Our study aimed at comparing three different growth models (the von Bertalanffy, the Robertson (logistic), and Gompertz) and also the absolute and relative growth characteristics of this species. Our results show that the relationship between the age of the long-nose skate and its total length is adequately explained by the Robertson (logistic) growth model, with the Gompertz growth model being the second best.

**Abstract:**

This study aims to determine the age and growth characteristics of *Dipturus oxyrinchus* living in the Northeastern Mediterranean Sea and to present data that can provide a comparison with previous studies on the same subject. A total of 255 long-nose skates at a total length of 12.2–93.5 cm and weight of 8.34–3828 g were collected as non-target species from a commercial fishing boat. The male−female ratio was determined as 1:1.27. Using the von Bertalanffy equation and the Gompertz or logistic growth models, the growth parameters of *Dipturus oxyrinchus* were estimated as L∞ = 154.0, K = 0.064, t_0_ = −1.622; L∞ = 104.0, K = 0.35, *I* = 4.99; L∞ = 128.40, K = 0.19, *I* = 4.39 for all individuals, respectively. Maximum absolute growth was calculated as 9.33 cm at 5–6 years of age. Maximum relative growth at 1–2 years of age was estimated as 36.39%. Both absolute and relative growth were minimal in the 11–12 age group. The highest condition factor value was estimated as 0.416 in the 8-year-old group. As a result, the growth data of long-nose skates were obtained for the first time in the Northeastern Mediterranean Sea.

## 1. Introduction

The long-nosed skate, *Dipturus oxyrinchus* (L. 1758), is a demersal species that lives at depths of 70−1230 m on sandy or muddy substrates and, rarely, on rocky and pebbly grounds; it is usually found at depths of 200−500 m [1,2,3,4,5,6]. The long-nosed skate ranges from Norway to Senegal, from the Northeast Atlantic to the Faroe Islands, Skagerrak (the strait connecting the Baltic Sea to the North Sea), the Canary Islands, the Madeira Islands and the Mediterranean Sea [3] and has little commercial value [4]. Sizes between 60 and 100 cm are common, but the largest recorded individual was 150 cm [1]. *Dipturus oxyrinchus* is globally recognized as a Near Threatened (NT) species by the International Union for Conservation of Nature (IUCN) [7]. Long-nosed skates have been studied satisfactorily by researchers during recent years in other parts of the Mediterranean Sea in terms of age, growth, distribution, systematics, length–weight relationships (LWR) and feeding habits. However, no data on growth parameters are available for this species in the Northeastern Mediterranean Sea [8,9,10,11,12]. The presence of juveniles and mature males of the long-nosed skate was previously reported by Başusta and Başusta [13] from the same region. Griffiths et al. [14] concluded that Mediterranean long-nosed skates may have been genetically isolated from other stocks (e.g., Atlantic). This study aimed to determine the growth characteristics of long-nosed skates living in the Northeastern Mediterranean Sea and compare them with the data reported in other studies conducted in the same region.

## 2. Materials and Methods

### 2.1. Collecting of Samples

The long-nosed skate individuals were caught by a commercial trawler (F/V NIHAT BABA/31-A-1463) in Iskenderun Bay (Figure 1; 36°29′200′′ N; 35°05′973′′ E–36°07′052′′ N; 35°17′936′′ E–36°07′148′′ N; 35°17′978′′ E–36°13′720′′ N; 35°22′998′′ E–36°13′650′′ N; 35°23′032′′ N–36°16′622′′ E; 35°18′509′′ N) between May 2015 and June 2016. The samples were collected monthly. The bottom trawl gear used was equipped with a 42 mm stretched-mesh size net at the cod-end. Each hauling lasted three hours, with a trawling speed of 2.2–2.9 knots. Approximate sampling depths ranged between 100–150 m, 150–200 m and 200–400 m. After fishing, all samples were transported with ice to the laboratory of the Fisheries Faculty, Firat University. Total length (*L*) was measured as a straight line from the tip of the rostrum to the end of the tail to the nearest mm and body mass (*W*) was weighed with a 1 g accuracy for each individual.

### 2.2. Processing of Vertebral Centra

A section of 10–12 vertebral centra was removed from the widest portion of the body of 255 long-nosed skate specimens (143 females and 112 males) and subsequently labeled, frozen and stored until further processing. Vertebrae were later thawed and cleaned of excess tissue, rinsed in tap water and then stored in a 70% ethanol solution. Three random vertebrae from each sample were removed from the ethanol and air-dried [15,16]. Smaller centra of less than 5 mm were fixed to a clear glass slide using resin (Crystol bond 509™) and were sanded with a Dremel™ tool to replicate a sagittal cut [17,18,19]. Vertebral sections (0.6 mm thick) were taken using a Ray Tech™ (Littleton, CO, USA) gam saw for large centra >5 mm in diameter [20]. Vertebral cross-sections were mounted on microscope slides using clear resin (Cytoseal 60; Fisher Scientific, Pittsburgh, PA, USA) [21].

### 2.3. Age Assessment and Verification

Vertebral cross-sections were examined under a Leica S8 APO™ (Singapore) microscope using LAS software (Version 4.8.0, Leica Microsystems Limited, Heerbrugg, Switzerland). One growth band was defined as an opaque and translucent band pair that traversed the intermedialia and clearly extended into the corpus calcareum (Figure 2) [22,23,24].

Each vertebral cross section was examined by two readers (reader 1 = NB and reader 2 = FVO). Reader 1 made two nonconsecutive band-counts of sampled vertebral cross-sections without prior information of the long-nosed skate’s length or former counts. Reader 2 made two consecutive counts from 50 randomly selected vertebrae sections. Vertebral cross-sections that had an instability of more than two years between each reading were eliminated from further analyses. Count reproducibility was compared by the percent agreement (%PA) and coefficient of variation (%CV) [25], as well as the index of average percent error between the readers (%IAPE) [26]. All were determined using the following mathematical equations:(1)$$\mathrm{PA}=\frac{No.\;agreed}{No.\;read}\times 100$$
(2)$${\mathrm{CV}}_{\;j\;}=100\times \frac{\sqrt{\sum_{i=1}^{R}\frac{{\left(X_{ij}-X_{j}\right)}^{2}}{R-1}}}{x_{j}}\;$$
(3)$$\mathrm{IAPE}=\frac{1}{R}\;\sum_{i=1\;}^{R}\frac{\left|x_{ij}-x_{j}\right|}{x_{j}}\times 100$$
where *R* is the number of readings; X*_ij_* is the count from the *_j_*th fish at the *i*th reading and *Xj* is the mean age calculated for the *j*th fish from *i* readings.

Pair-wise age-reader comparisons were independently generated by the two readers by making nonconsecutive band counts from a random sample of 50 vertebral sections [27]. All statistical tests were performed with R software version 4.1 [28], with a significance level set at 5%.

### 2.4. Total Length–Weight Relationship and Growth Modeling

The total length–weight relationship parameters of long-nosed skate were estimated according to the equation given below [29]:(4)$$W=aL^{b}$$
where *L* = total length (cm); *W* = body mass (g); *a* is a constant of proportionality and *b* is the allometric factor. The deviance of the estimated *b* values for long-nosed skate from the hypothetical value of 3 (i.e., isometric growth) was tested by a *t*-test at the 0.01 significance level [29,30].

The absolute length growth (ALG) and the relative length growth (RLG) of long-nosed skate were calculated as:(5)ALG =(Lt+∆t−Lt)/∆t
(6)RLG =(Lt+∆t−Lt/Lt)×100
where *L_t_* is total length at the start of the time interval and *L_t_*_+*∆t*_ is total length at the end of the time interval (*∆t*) [30,31].

The values of condition factor were obtained with the formula:(7)$$Kn=\left(\frac{W}{Lb}\right)\times 100$$
where *W*, *L* and *b* are as defined above [32].

The observed length-at-age data of the long-nosed skate were used as the dependent variable and the age as the independent variable, with the three most-used models to describe the growth of fish. The first model employed was the von Bertalanffy [33] growth equation (VBGM). The VBGM was formulated by Beverton and Holt [34] as:(8)$$L_{t}=L_{\infty }(1-e^{-K(t-t_{0})}),$$
where *L_t_* is the expected total length at age *t* years; *L*_∞_ is the asymptotic total length; *K* is the growth coefficient or curvature parameter indicating the rate at which long-nosed skates grow toward their *L*_∞_ and *t*_0_ is the theoretical age at zero total length.

The second growth model used was that of Gompertz [35], an S-shaped growth model (GGM) [36,37,38]:(9)$$L_{t}=L_{\infty }e^{-e^{-G(t-t_{i})}},$$
where *t*, *L_t_* and *L*_∞_ are the same as in the VBGM; *t_i_* is the age at the inflection point of the growth curve, i.e., the age at which the absolute growth rate starts to decrease and *G* is the instantaneous growth rate coefficient at age *t*_i_, where growth becomes asymmetrical.

The Robertson (logistic) model was also used as the last growth model. While *t*, *t*_i_*, L_t_* and *L*_∞_ are the same as in the previous models, *K* is a parameter that affects the rate of exponential growth:(10)$$Lt=\frac{L_{\infty }}{\left(1+e\left(-K\left(t-t_{i}\right)\right)\right)}$$

Akaike’s information criterion (AIC) was used to compare the different growth models. A smaller value of the AIC indicates that the observed data are closer to the fitted model [29,39]. The AIC is defined as −2 times the maximum value of the log likelihood ($\hat{L}$) plus 2 times the number of parameters (*p*) in the model including the estimated variance [40]:(11)$$\mathrm{AIC}=-2\ln(\hat{L})+2p$$

## 3. Results

### 3.1. Sample Composition, Sex and Vertebral Analysis

A total of 255 long-nosed skates ranging from 12.2 to 93.5 cm in total length and 8.34–3828 g in weight were collected as bycatch or discard species from a commercial fishing vessel between May 2015 and June 2016 (Table 1). The sex ratio (M/F) was determined as 1:1.27. The ratio of females to males was not statistically different from the expected 1:1 ratio between sexes (*p* > 0.05). The size frequency based on total length and age group is presented in Figure 3.

### 3.2. Age Estimation, Reading Precision and Age Bias

In this study, estimated ages ranged from 0 to 12 for females and from 0 to 9 for males. Age estimation for males and females by two independent readers did not show a considerable variation (Figure 4a,b). The highest PA with lowest IAPE and CV were found for males than female (Table 2).

### 3.3. Length–Weight Relationships, Growth Patterns and Condition Factor

#### 3.3.1. Length–Weight Relationships

The total length–weight relationships of *Dipturus oxyrinchus* for female, male and overall are presented in Figure 5. According to these results, positive allometric growth (*b* > 3) was demonstrated for the overall category. Regression results showed that the length and weight of the long-nosed skate was predicted significantly (r = 0.987, r^2^ = 0.974, F1,242 = 9092.525 *p* < 0.001) in all sexes. It is possible to state that 97% of the increase is due to that of the size of *Dipturus oxyrinchus* in all individuals for the present study. In addition, when *t*-test results related to the significance of the regression coefficients were analyzed (*t*-test = 95.355 *p* < 0.01), the proportion of the individual’s length to weight was found to be important.

#### 3.3.2. Growth Characteristics

The growth parameters were estimated for overall and each sex separately from the VBGM, the Robertson (logistic) and Gompertz growth model using nonlinear regression analysis, as presented in Table 3 and Figure 6. The relationship between total length and age was adequately described by the Robertson (logistic) growth model followed by the Gompertz growth model. The von Bertalanffy model performed relatively weakly compared to other growth models based on AIC values.

#### 3.3.3. The Relative and Absolute Growth Rates and Condition Factor

The maximum absolute growth was estimated as 9.33 cm with 5–6 years of age. The maximum relative growth was calculated as 36.39% with 1–2 years of age. Both in absolute and relative growth were observed as minimum in the 11–12 age group (Table 4). Average condition factor value of population was estimated as 0.363. The highest and lowest condition factor values were estimated as 0.416 in age group 8 and 0.308 in age group 2, respectively (Table 5).

## 4. Discussion

The overall length ranges recorded in our study were found to be smaller than those reported by Alkusairy and Saad [11] for the same species in Syrian waters (34.1–100 cm for females and 34.5–81.6 cm for males). Yigin and Işmen [8] reported a total length of 14.9–100 cm in females in Saros Bay (the North Aegean Sea), while Bellodi et al. [12] reported these values as 15.2–86.5 cm in males and 10.4–117.5 cm in females in Sardinian waters. Finally, Kadri et al. [9] reported 16.5–105 cm in females and 15.5–95 cm in males in the Gulf of Gabès (Southern Tunisia, Central Mediterranean). These values are very close to the values we found in our study. The Robertson (logistic) growth model estimates indicated that the long-nosed skate showed sexual dimorphism with females larger than males. These observations are consistent with other studies for *Dipturus oxyrinchus* in the Mediterranean Sea (e.g., Yigin and Işmen [8] in the North Aegean Sea, Kadri et al. [9] in Southern Tunisia and Bellodi et al. in Sardinian waters [12]). Sexual dimorphism appears to be a common feature for the Rajidae (e.g., blonde ray *Raja brachyura* [41] and thornback ray *Raja clavata* [42]). The percentage of females and males for all samples was 56.07% and 43.93%, respectively. This was not statistically different from the expected 1:1 ratio between the genders. All genders were equally distributed confirming the pattern proposed by Yigin and Işmen [8], Kadri et al. [9] and Bellodi et al. [12]. Growth bands were highly legible and visible in cross-sections, with an easily recognizable birthmark. No staining technique was used to determine the age of *Dipturus oxyrinchus*, with a maximum age of 12 noted in females and 9 in males. Differences in age determination after age 9 for both sexes are related to the small number of individuals being sampled. The age estimation process ensured a high level of count repeatability among readers (IAPE = 0.53%; %CV = 0.75; %PA = 90.98) and no signs sensitive to bias were detected among readers. These precision values are acceptable [43]. The b parameter of the length–weight relationship of *Dipturus oxyrinchus* showed positive allometric growth for both genders in the current study. The estimated *b* values for *Dipturus oxyrinchus* by region are shown in Table 6, which are very close to our study’s findings. Other b values were reported as 3.539 for the south coasts of Portugal by Borges et al. [44] and 3.40 for North Aegean Sea by Filiz and Bilge [45]. These values differ from those of our study, which may be due to the small sample size of the fish or the fact that samples were made in different seasons. Relative and absolute growth rates and condition factor for *Dipturus oxyrinchus* were calculated for the first time in our study and therefore no comparison with other studies could be made. Absolute growth rates indicate actual growth between two years (ages) in terms of weight or length. Absolute growth rate decreases with age (t) and provides information about the years (ages) when growth is highest. The way the absolute growth rate is calculated depends strongly on the size the fish has reached. For comparison purposes, the relative growth rate may be more useful. This is used to determine age-related growth rate in natural populations [30].

Bellodi et al. [12] emphasized that the Gompertz function provided the best fit among the four growth models examined. Additionally, Liu et al. [49] stated that multiple model applications should be tested in elasmobranch age and growth studies. They also indicated that the Robertson (logistic) and Gompertz models provide the best fit for small-sized demersal skates/rays living in deep water. In our study, according to the AIC values, the logistic and Gompertz models were found to be more appropriate in describing the growth parameters of *Dipturus oxyrinchus*. These results agree with Liu et al. [49] and Bellodi et al. [12]. The logistic parameters determined in our study show that females attain a slightly larger asymptotic TL∞ (103.54 cm) than males (103.23 cm). In addition, the K values of the long-nosed skate were found to be similar for both genders. These growth rates appear to be similar with other skate species of similar size in the Mediterranean Sea. Bellodi et al. [12] suggested that the best growth model was the Gompertz model and estimated L∞ as 127.55 cm for all genders. This result is very close to the asymptotic value (L∞ = 128.40 cm) calculated with the Gompertz model for all individuals in our study. Yigin and Işmen [8] reported that, for both genders, the L∞ was 256.46 cm and the K was 0.04, the t_0_ value was −1.17 year and the maximum age was 9 years. The above authors [8] found individuals aged up to 9 years and a total length of up to 100 cm in their study. The largest individual captured in our study (93.5 cm) was smaller than that reported by the above researchers, despite being older. This shows that there may be a mistake in reading the older age rings. This leads to an overcalculation of the asymptotic length. For females, Kadri et al. [9] reported the L∞ as 123.9 cm, the K as 0.08, the t_0_ as −1.26 and the maximum age as 25; for males, the L∞ as 102.1 cm, the K as 0.12, the t_0_ as −1.18 and the maximum age as 22. These researchers found that the oldest individual was 105 cm in total length and 25 years old. Again, these values were very high compared to those in our study, which therefore indicated to us that mistakes might have been made in the reading of the age rings of the older individuals. These errors may have caused an underestimation of the L∞ value. Considering that the largest *Dipturus oxyrinchus* caught in nature is 150 cm in TL, the L∞ value should not be lower than this value.

## 5. Conclusions

Our work ensured basic growth parameters and the best fit among the three growth models for the long-nosed skate and that the Robertson (logistic) growth model was the best model to describe the species growth. Results of the research showed that the long-nosed skate has life history features similar to other Rajidae species in the Mediterranean Sea and are long-lived and slow growing. The present study has provided the first analysis of growth characteristics and data for *Dipturus oxyrinchus* for conservative management plans in the Northeastern Mediterranean Sea.

## Figures and Tables

**Figure 1 animals-12-03443-f001:**
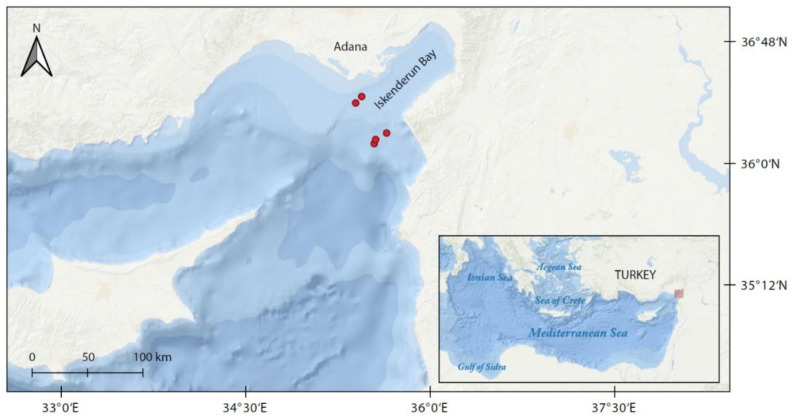
Red dots indicate out of Iskenderun Bay, Turkey, where *Dipturus oxyrinchus* specimens were collected.

**Figure 2 animals-12-03443-f002:**
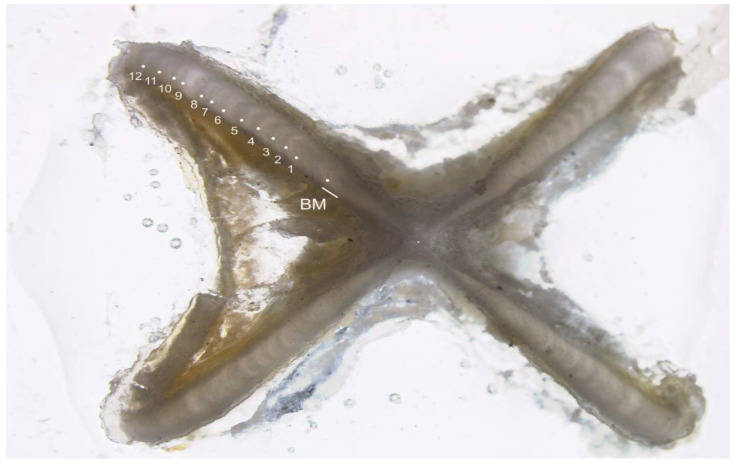
A vertebral cross-section of an estimated 12-year-old *Dipturus oxyrinchus* (total length = 93.5 cm, female) (BM, Birth Mark). White dots indicate opaque bands.

**Figure 3 animals-12-03443-f003:**
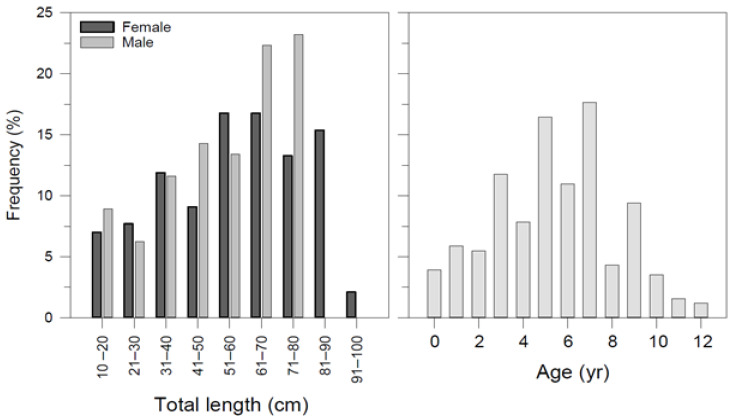
Frequency distribution of *Dipturus oxyrinchus* inhabiting the Northeastern Mediterranean Sea, according to their total length and age.

**Figure 4 animals-12-03443-f004:**
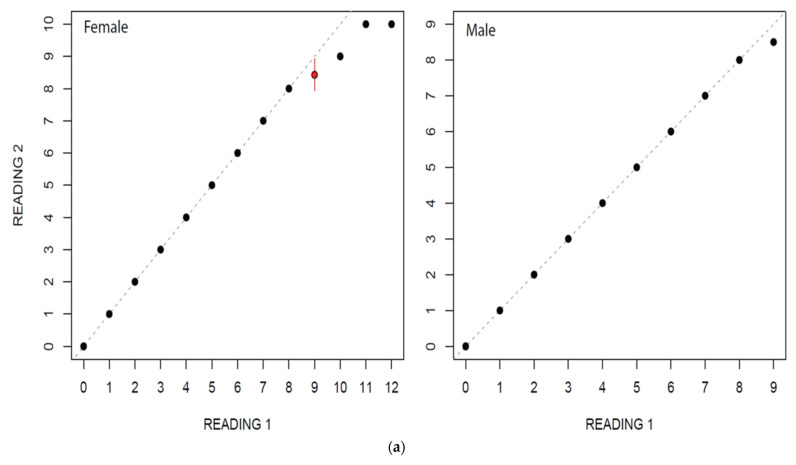
(**a**) Age bias graphs for *Dipturus oxyrinchus* inhabiting the Northeastern Mediterranean Sea. (**b**) Age bias plots for two readers for aging *Dipturus oxyrinchus* inhabiting the Northeastern Mediterranean Sea. Plots illustrate reference reader’s age estimates on the *x*-axis; the mean difference (circles) and distribution of the differences between corresponding ages (vertical lines) are represented on the *y*-axis.

**Figure 5 animals-12-03443-f005:**
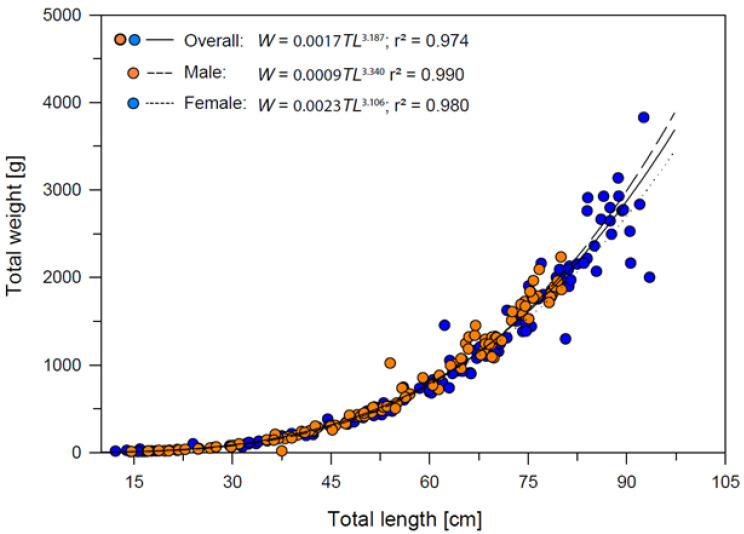
Total length−weight relationships of *Dipturus oxyrinchus* inhabiting the Northeastern Mediterranean Sea.

**Figure 6 animals-12-03443-f006:**
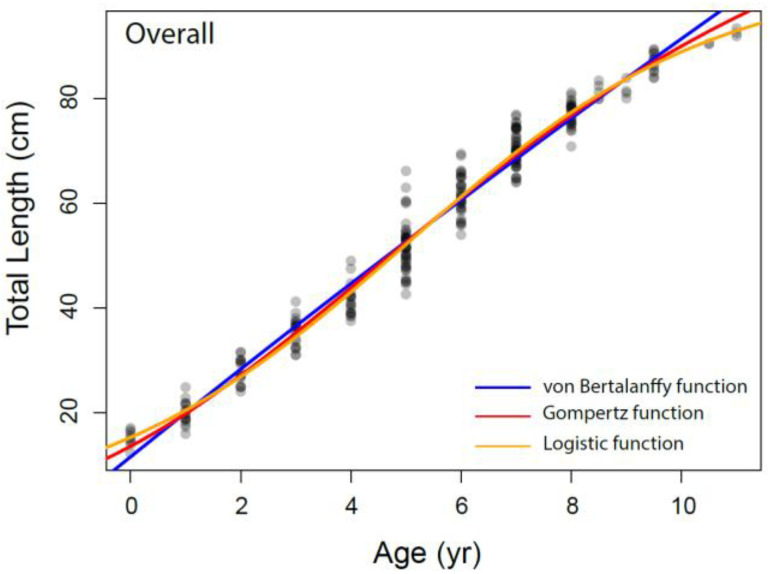
The different growth models fitted to the data of *Dipturus oxyrinchus* inhabiting the Northeastern Mediterranean Sea.

**Table 1 animals-12-03443-t001:** Descriptive statistics for *Dipturus oxyrinchus* inhabiting the Northeastern Mediterranean Sea.

Group	*n*	Total Weight (g)		Total Length (TL, cm)	
		Mean SE	Min–Max	Mean SE	Min–Max
Female	143	1000.28 ± 902.78	8.50–3828.00	57.38 ± 22.05	12.20–93.50
Male	112	825.44 ± 666.55	8.34–2234.00	53.97± 19.24	14.60–80.10

SE: Standard Error; Min: Minimum; Max: Maximum.

**Table 2 animals-12-03443-t002:** Summary statistics for the coefficient of variation (CV), percentage of agreement (%PA) and index of average percentage error (IAPE) to determine the precision of age readings of *Dipturus oxyrinchus* inhabiting the Northeastern Mediterranean Sea.

Group	Readers	*n*	R	CV	PA
Overall	Reader 1 vs. Reader 2	255	2	0.75	90.98
Female	Reader 1 vs. Reader 2	143	2	1.27	84.62
Male	Reader 1 vs. Reader 2	112	2	0.74	99.11

*n* = sample size and R = number of readings.

**Table 3 animals-12-03443-t003:** The growth parameters for *Dipturus oxyrinchus* inhabiting the Northeastern Mediterranean Sea derived from different growth models.

Group	Growth Models	*L_∞_* (cm)	*K* (year^−1^)	*t_0_* (year)	*I* (year)	*AIC*
Overall	von Bertalanffy	154.01	0.06	−1.62		1401.60
	Logistic	104.14	0.35		4.99	1350.69
	Gompertz	128.40	0.19		4.39	1363.98
Female	von Bertalanffy	152.55	0.06	−1.62		788.90
	Logistic	103.54	0.36		4.91	751.83
	Gompertz	124.73	0.19		4.19	761.12
Male	von Bertalanffy	161.73	0.06	−1.63		788.90
	Logistic	103.23	0.35		4.98	595.02
	Gompertz	141.44	0.17		5.02	598.61

*L*_∞_ = the asymptotic length (cm); *K* = the growth rate (year^−1^); *t*_0_ = the time when *L* = 0 (year); *I* = the age at the inflection point; and AIC = Akaike Information Criterion.

**Table 4 animals-12-03443-t004:** The relative and absolute growth rates of for *Dipturus oxyrinchus* inhabiting the Northeastern Mediterranean Sea.

	Absolute Growth Rate			Relative Growth Rate		
Age Group	Overall	Female	Male	Overall	Female	Male
0–1	4.94	5.14	5.22	32.59	33.72	35.17
1–2	7.31	6.74	7.73	36.39	33.06	38.5
2–3	8.86	8.32	9.56	32.3	30.7	34.39
3–4	8.71	9.95	9.13	24.01	28.07	24.42
4–5	9.09	11.53	9.67	20.2	25.4	20.8
5–6	9.33	7.7	9.33	17.26	13.53	16.62
6–7	7.54	6.28	5.48	11.9	9.72	8.36
7–8	5.2	5.39	4.82	7.33	7.6	6.79
8–9	4.23	4.97	3.1	5.56	6.52	4.09
9–10	6.67	5.78	-	8.3	7.11	-
10–11	2.94	2.94	-	3.38	3.38	-
11–12	2.73	2.73	-	3.03	3.03	-

**Table 5 animals-12-03443-t005:** The condition factor for *Dipturus oxyrinchus* inhabiting the Northeastern Mediterranean Sea.

	ALL		FEMALE		MALE	
AGE	*N*	K	*N*	K	*N*	K
0	10	0.409	8	0.454	2	0.273
1	15	0.317	4	0.221	11	0.279
2	14	0.308	9	0.377	5	0.267
3	30	0.311	17	0.298	13	0.303
4	20	0.335	5	0.25	15	0.349
5	42	0.36	24	0.268	18	0.38
6	28	0.375	18	0.32	10	0.416
7	45	0.383	24	0.365	21	0.389
8	11	0.416	3	0.404	8	0.418
9	24	0.386	15	0.389	9	0.386
10	9	0.405	9	0.411	-	-
11	4	0.351	4	0.322	-	-
12	3	0.362	3	0.362	-	-
		0.363 ± 0.037		0.342 ± 0.060		0.346 ± 0.060

**Table 6 animals-12-03443-t006:** Length–weight relationship values for *Dipturus oxyrinchus* from different regions.

	Sexes	*n*	L_min-max_ (cm)	W_min-max_ (g)	a	b	r^2^	Researchers
South coasts, Portugal	Σ	8	30.2–55.4	88.0–702.7	0.00048	3.539	0.99	Borges vd. [44]
North Aegean Sea	Σ	8	17.9–62.2	10.44–850.48	0.0007	3.40	0.99	Filiz and Bilge [45]
South of Sicily and South of Malta	Σ	-	23.0–124.0	-	0.00128	3.250	-	Geraci et al. [46]
Northwestern Mediterranean Sea	Σ	2	27.4–33.6	58–84	0.0025	3	-	Barría et al. [47]
Gulf of Saros, Turkey	Σ	118	10.0–63.2	9.0–4056.0	0.00423	3.291	0.998	Işmen et al. [48]
Sardinian waters, Italy	♀	531	10.9–115.5	-	0.0012	3.2498	0.98	Bellodi et al. [12]
	♂	448	14.7–101.5	-	0.0009	3.327	0.99	
Gulf of Saros, Turkey	Σ	179	14.9–100.0	8.0–4047.0	0.00083	3.35	0.996	Yigin and Işmen [8]
	♀	89	14.9–100.0	8.0–4047.0	0.00077	3.37	0.997	
	♂	90	15.2–86.5	8.0–2510.0	0.00088	3.34	0.996	
Mersin Bay, Northeastern Mediterranean Turkey	Σ	255	12.20–93.50	8.34–3828.0	0.0017	3.187	0.974	In this study
	♀	143	12.20–93.50	8.50–3828.0	0.0023	3.106	0.990	
	♂	112	14.60–80.10	8.34–2234.0	0.0009	3.340	0.980	

*n*: sample size, L_min-max_: minimum-maximum length, W_min-max_: minimum-maximum weight, a: intercept, b: slope of the equation, r^2^: coefficient of determination.

## Data Availability

Data associated with this research are available and can be obtained by contacting the corresponding author.

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
