# Peer review of "Growth Characteristics of Long-Nosed Skate Dipturus oxyrinchus (Linnaeus, 1758) Inhabiting the Northeastern Mediterranean Sea"

_animals, 2022, doi:10.3390/ani12233443_

Round 1

Reviewer 1 Report

-Line 3: I would recommend adding ”Sea“ at the end of the title.

-General comment: English language quality must be improved throughout the manuscript.

-Line 35: The depth range is not correct, please check Weigmann (2016: Annotated checklist of the living sharks, batoids and chimaeras (Chondrichthyes) of the world, with a focus on biogeographical diversity. Journal of Fish Biology 88(3), 837–1037. https://doi.org/10.1111/jfb.12874).

-Lines 35–37: It is not clear what the authors try to express and a citation is also missing. This sentence must be rewritten.

-Lines 37–40: This sentence must also be rewritten. Furthermore, the information provided is not correct as Dipturus oxyrinchus has not been reported from the Northwest Atlantic. For up-to-date information on the distribution of D. oxyrinchus please see Last PR, Séret B, Stehmann MFW, Weigmann S (2016) Skates, family Rajidae. In: Last PR, White WT, de Carvalho MR, Séret B, Stehmann MFW, Naylor GJP (eds) Rays of the world. CSIRO Publishing, Melbourne, pp 204–363.

-Lines 40–41 and lines 42–46: Further example of sentences requiring language improvement.

-Lines 46 and 49: Be consistent with writing northeastern / Northeastern and with using Sea or not throughout the manuscript including the title and abstract.

-Lines 56–58: Difficult to read with all the parentheses, please rephrase and use n-dash for all ranges (ranges need to be checked throughout the manuscript, e.g. line 159).

-Lines 94–96: What about Reader 2? Detailed description is only provided for Reader 1.

-Lines 159–160: First sentence says that there were more females than males but the sex ratio indicates that there were more males. This needs to be checked and corrected (possibly also in the Abstract).

-Table 1: Minimum weights of 8.5 g (in females) and 8.34 g (in males) appear to be erroneous. Can you please check if possibly 85 and 83.4 g are the correct values?

-Lines 192–194: This sentence does not make sense, must be rewritten.

-Lines 195–198: This sentence is another example why the language quality must be improved throughout the manuscript.

-Discussion: The discussion should be rewritten extensively. Furthermore, additional aspects should be discussed, e.g. the finding that the asymptotic lengths calculated in the present study tend to be larger for males, which is in contrast to the general observation that females grow larger than males in many skate species. A more detailed and critical discussion of the results of the present study with available other studies is also desirable. Additionally, the significance of the results of the present study should be discussed, e.g. related to fishery, management and population status.

-Conclusions: Similar to the Discussion, the Conclusions also need to be rewritten to be more concise and interpret the present results in a larger context.

Author Response

Dear Reviewer,

Thank you very much for your corrections and comment on the article. All the corrections were done. you can find the corrections on the text.

-Line 3: I would recommend adding ”Sea“ at the end of the title.

 It was added

-General comment: English language quality must be improved throughout the manuscript.

 Done

-Line 35: The depth range is not correct, please check Weigmann (2016: Annotated checklist of the living sharks, batoids and chimaeras (Chondrichthyes) of the world, with a focus on biogeographical diversity. Journal of Fish Biology 88(3), 837–1037. https://doi.org/10.1111/jfb.12874).

 It was changed and literature added

-Lines 35–37: It is not clear what the authors try to express and a citation is also missing. This sentence must be rewritten.

 This sentence has been rewritten

-Lines 37–40: This sentence must also be rewritten. Furthermore, the information provided is not correct as Dipturus oxyrinchus has not been reported from the Northwest Atlantic. For up-to-date information on the distribution of D. oxyrinchus please see Last PR, Séret B, Stehmann MFW, Weigmann S (2016) Skates, family Rajidae. In: Last PR, White WT, de Carvalho MR, Séret B, Stehmann MFW, Naylor GJP (eds) Rays of the world. CSIRO Publishing, Melbourne, pp 204–363.

 It was corrected and literature added

-Lines 40–41 and lines 42–46: Further example of sentences requiring language improvement.

 Done

-Lines 46 and 49: Be consistent with writing northeastern / Northeastern and with using Sea or not throughout the manuscript including the title and abstract.

 All the “northeastern” were changed as “Northeastern” and with using Sea throughout the manuscript

-Lines 56–58: Difficult to read with all the parentheses, please rephrase and use n-dash for all ranges (ranges need to be checked throughout the manuscript, e.g. line 159).

 We used n-dash for all ranges

-Lines 94–96: What about Reader 2? Detailed description is only provided for Reader 1.

Reader 2 made two consecutive counts from 50 randomly selected vertebrae sections. This sentence has been added.

-Lines 159–160: First sentence says that there were more females than males but the sex ratio indicates that there were more males. This needs to be checked and corrected (possibly also in the Abstract).

 Corrected in the Abstract

-Table 1: Minimum weights of 8.5 g (in females) and 8.34 g (in males) appear to be erroneous. Can you please check if possibly 85 and 83.4 g are the correct values?

This data has been checked by me, there is no erroneous.

 -Lines 192–194: This sentence does not make sense, must be rewritten.

 This sentence has been rewritten

-Lines 195–198: This sentence is another example why the language quality must be improved throughout the manuscript.

 This sentence has been corrected

-Discussion: The discussion should be rewritten extensively. Furthermore, additional aspects should be discussed, e.g. the finding that the asymptotic lengths calculated in the present study tend to be larger for males, which is in contrast to the general observation that females grow larger than males in many skate species. A more detailed and critical discussion of the results of the present study with available other studies is also desirable. Additionally, the significance of the results of the present study should be discussed, e.g. related to fishery, management and population status.

 The discussion has been rewritten

-Conclusions: Similar to the Discussion, the Conclusions also need to be rewritten to be more concise and interpret the present results in a larger context.

  Done

Reviewer 2 Report

·      The aim of this study is to identify the age and growth characteristics of Dipturus oxyrinchus living in the Eastern Mediterranean Sea and to present data that can provide a comparison with previous studies on the same topic. This sort of work has been done before by multiple researchers. even though this work is important, the processing and the calculation are very good and presented in a readable and nice configuration.  I think that it will be better to present this study as a mini-review because very similar results have already been presented in previous studies (which were done with very similar methods)

·      If the authors want to emphasize a connection to the preservation of the marine environment, it is better that they use the article of Griffiths et al. (2011) whose conclusion was, based on egg capsules and adult sizes, that longnose skates in the Mediterranean may be genetically isolated from other stocks (Atlantic). This result has important conservation implications for the threatened longnose skate and conservation efforts are needed in the Mediterranean Sea.

Author Response

Dear Reviewer,

Thank you very much for your corrections and comments on the article. Griffiths et al. (2011) and some references have been added to the Introduction. We have no Atlantic forms, and cannot compare at this time. However, we may suggest that such studies be carried out in the future. You can find all the corrections on the article text.

Reviewer 3 Report

This study focuses on the “Growth characteristics of long-nosed skate Dipturus oxyrinchus (Linnaeus, 1758) inhabiting the Northeastern Mediterranean”. This is a very interesting topic for the species and study area. However, the results are not presented in a sufficient way and they are not properly discussed, particularly their implications. Despite the importance of such a study, all sections have serious issues that should be addressed before this draft is considered for publication.

My comments are described below in major and minor comments.

Major comments

The authors should also rephrase almost all text. The English standards are very poor at a state where this manuscript should not be accepted for publication.

The tile should also change. Firstly, Iskenderun bay is not in the Northeastern Mediterranean and secondly, samples from one bay are not be representative of such a large area, particularly as the authors have proven that samples from different areas exhibit different life history traits. You should therefore replace northeastern Mediterranean with a more appropriate term throughout the text.

The authors do not fully discuss the potential of their results and the implications they have to the conservation and management of the species. For example, this skate is caught as bycatch, but does it have any commercial value in Turkey?

Minor comments

Simple summary and Abstract

The English standards are poor and these sections should be rephrased.

Introduction

Lines 33-35: This sentence should be rephrased. There are verbs missing in parts of the sentence, prepositions not correctly used.

Line 36: What do the authors mean with “large scale living species”? And what about the “

 with the widest geographical distribution”? There are species that have wider distribution than D. oxyrinchus.

Line 37: You should start a sentence with the full name of the species.

Line 37-51: The whole section should be rephrased; there are many grammar and syntax mistakes.

I find the Introduction a bit poor as it is only 18 lines.

Material and methods

From this point forward, I will not focus on the English standard of the draft and I will only comment on issues regarding the quality of the study.

Line 62: Did the authors actually use “dry ice” or did they mean something else?

Line 71: Replace “A section of 10-12 vertebral centra were” with “A section of 10-12 vertebral centra was”.

Lines 106-107: What do the authors mean with the work “readings”?

Lines 110-111: Why did the authors use two significance levels (5% and 1%)?

Lines 122-127: The authors should present the ALG, RLG and Kn relationships with the similar was as in the previous equations.

Lines 144-145: What about the K in the Robertson model?

What about the ethical approval used for this work? Did they authors manage to get one? It is found in the acknowledgments but it should also be stated at this point.

Results

Lines 159-161: Was there any statistical significance for the sex ration?

Table 1: What does the SE represents? There is no explanation on the legend.

Figure 5 is presented before Figure 4. This should change. We first report Figure 4 and then Figure 5.

Table 2; What is “R”? No explanation is presented on the legend.

Section 3.2: What about the differences in age determination after year 9 for both sexes? The authors did not mention anything about it.

Lines 190-191: I cannot understand why there are two TL-TW relationships for females. I assume one must be the relationship for both sexes?

Table 3: What is “I”? No explanation is presented on the legend.

Discussion

Line 233: Why did the authors not use any staining? Have they tried it and decided that the results are worse than those without staining? If not, did they use a study that proved that non-staining is better than staining for this species? If they did, then they should mention it.

Line 234: What do the authors mean with “rather”? Maybe staining would have been better at reading growth bands?

Lines 238-239: The authors move to the TW-TL relationship without providing any information about the change of subject. The readers need to go to Table 6 in order to realise what the authors are describing/discussing. Please rephrase and provide more details about this relationship.

Lines 242-243: Whose sample size? This study or the rest? What about the potential seasonal difference? Any chance the samples from the other studied were collected in different season?

Line 259: What do the authors mean with “future ages”?

Lines 253-268: These lines should be rephrased. The statements are not fully described, the results of the study are not fully presented. What about the L∞ of this study? How does this compare with previous studies?

Conclusions

Lines 272-274: I am afraid that this sentence states the obvious. Of course D. oxyrinchus will have similar life history characteristics as other Rajidae. But from where? The Mediterranean Sea? Dipturus from other seas?

Author Response

Dear Reviewer,

Thank you very much for your corrections and comments on the article. All the corrections were done. you can find the corrections on the text.

Simple summary and Abstract

The English standards are poor and these sections should be rephrased.

 Done

Introduction

Lines 33-35: This sentence should be rephrased. There are verbs missing in parts of the sentence, prepositions not correctly used.

This sentence were corrected

Line 36: What do the authors mean with “large scale living species”? And what about the “

 with the widest geographical distribution”? There are species that have wider distribution than D. oxyrinchus.

This sentence were corrected

Line 37: You should start a sentence with the full name of the species.

The full name of the species has been added

Line 37-51: The whole section should be rephrased; there are many grammar and syntax mistakes.

I find the Introduction a bit poor as it is only 18 lines.

Some literature additions have been made to the introduction. 

Material and methods

From this point forward, I will not focus on the English standard of the draft and I will only comment on issues regarding the quality of the study.

Line 62: Did the authors actually use “dry ice” or did they mean something else?

it was written by mistake, “regional word”. We have corrected only as “ice“ in text.

Line 71: Replace “A section of 10-12 vertebral centra were” with “A section of 10-12 vertebral centra was”.

Done

Lines 106-107: What do the authors mean with the work “readings”?

“readings” it means “age readings”

Lines 110-111: Why did the authors use two significance levels (5% and 1%)?

it was written by mistake. The significance levels was used “5%”. It was corrected.

Lines 122-127: The authors should present the ALG, RLG and Kn relationships with the similar was as in the previous equations.

Done. See text.

Lines 144-145: What about the K in the Robertson model?

K is a parameter that affects the rate of exponential growth and added.

What about the ethical approval used for this work? Did they authors manage to get one? It is found in the acknowledgments but it should also be stated at this point.

There is an obligation to obtain ethical approval for studies conducted on animals in Turkey.

Results

Lines 159-161: Was there any statistical significance for the sex ration?

Ratio of females to males was not statistically different from the expected 1:1 ratio between the sexes (P > 0.05).

Table 1: What does the SE represents? There is no explanation on the legend.

“SE” it means Standart Error and added at the bottom of the table

Figure 5 is presented before Figure 4. This should change. We first report Figure 4 and then Figure 5.

Figures have been changed.

Table 2; What is “R”? No explanation is presented on the legend.

“R” is number of Readings and added at the bottom of the table

Section 3.2: What about the differences in age determination after year 9 for both sexes? The authors did not mention anything about it.

The differences in age determination after year 9 for both sexes are due to the small number of samples.

Lines 190-191: I cannot understand why there are two TL-TW relationships for females. I assume one must be the relationship for both sexes?

This sentence has been corrected. First TL-TW relationship is for both sexes

Table 3: What is “I”? No explanation is presented on the legend.

 I = the age at the inflection point

Discussion

Line 233: Why did the authors not use any staining? Have they tried it and decided that the results are worse than those without staining? If not, did they use a study that proved that non-staining is better than staining for this species? If they did, then they should mention it.

The growth bands were readable and visible on the cross-sections and it had an easily recognizable birthmark. Therefore, no staining method was used.

Line 234: What do the authors mean with “rather”? Maybe staining would have been better at reading growth bands?

We can easily read the growth bands all the skate and rays. Because we tried four staining methods for other skates. We decided it wasn't necessary for every skates. This word (rather) is a relative concept and it was deleted.

Lines 238-239: The authors move to the TW-TL relationship without providing any information about the change of subject. The readers need to go to Table 6 in order to realise what the authors are describing/discussing. Please rephrase and provide more details about this relationship.

Done

Lines 242-243: Whose sample size? This study or the rest? What about the potential seasonal difference? Any chance the samples from the other studied were collected in different season?

The samples were collected monthly in our study. This sentence was corrected. (This difference in other studies may be caused by the lower sample size or collected in different season).

Line 259: What do the authors mean with “future ages”?

in later age rings

Lines 253-268: These lines should be rephrased. The statements are not fully described, the results of the study are not fully presented. What about the L∞ of this study? How does this compare with previous studies?

Done

 Conclusions

Lines 272-274: I am afraid that this sentence states the obvious. Of course D. oxyrinchus will have similar life history characteristics as other Rajidae. But from where? The Mediterranean Sea? Dipturus from other seas?

“In the Mediterranean Sea” and it was added.

Round 2

Reviewer 1 Report

The authors provided a significantly improved manuscript, in which I only found one small mistake: in the footer to Table 1, it should be Standard instead of Standart.

Author Response

Dear Reviewer, 

Thank you very much for the correction. You can find it as attached file.

yours sincerelly

Reviewer 2 Report

Very nice improvment

Author Response

Dear Reviewer,

Thank you very much for your comments, you can find the MS as attached file.

yours sincerelly

Reviewer 3 Report

This is the second round of review for the study “Growth characteristics of long-nosed skate Dipturus oxyrinchus 2 (Linnaeus, 1758) inhabiting the Northeastern Mediterranean Sea”. The manuscript has improved following the suggestions from both reviewers. However, the English standard is not good and the authors should look for an editor.

Finally, there are few minor issues that should be addressed.

Introduction

Lines 35-36: I still do not consider that D. oxyrinchus is a species with a wide distribution. You should remove this sentence or provide references.

Line 36: You should start a sentence with the full name of the species. It has not been corrected. The same applies throughout the manuscript.

Lines 47-49: This is not what Griffiths et al., 2011 did. He did not use egg-capsules nor sizes. He used genetic data. You need to read the paper and not only the abstract.

On my first review, I have mentioned that the Introduction is relatively small. You need to add more than references (only 3 in your case) to improve the Introduction.

Methods

They should mention the ethical approval at the beginning of this section.

Results

Lines 170-171: What was the statistical difference for the sex ratio? You should mention it.

Lines 199-200: This sentence is redundant as you describe and mention Figure 5 in the next sentences.

Discussion

Lines 262-263: The significance in the sex ratio is a result. The authors should discuss what are the implications of the lack of significance.

Lines 270-272: This is a great result that this study is the first to calculate “relative and absolute growth rates”. But what does that mean? What information do we get from the rates?

Author Response

Dear Reviewer,

Thank you very much for your contructive comments and corrections. All the corrections were done on the text.

Introduction

Lines 35-36: I still do not consider that D. oxyrinchus is a species with a wide distribution. You should remove this sentence or provide references.

This sentence was removed

Line 36: You should start a sentence with the full name of the species. It has not been corrected. The same applies throughout the manuscript.

Done

Lines 47-49: This is not what Griffiths et al., 2011 did. He did not use egg-capsules nor sizes. He used genetic data. You need to read the paper and not only the abstract.

This sentence was corrected

On my first review, I have mentioned that the Introduction is relatively small. You need to add more than references (only 3 in your case) to improve the Introduction.

Since there are not many studies on Dipturus oxyrinchus, especially on growth, only three have been added.

Methods

They should mention the ethical approval at the beginning of this section.

Done

Results

Lines 170-171: What was the statistical difference for the sex ratio? You should mention it.

The ratio of females to males was not statistically different from the expected 1:1 ratio between the genders (P > 0.05). Done

Lines 199-200: This sentence is redundant as you describe and mention Figure 5 in the next sentences.

Deleted

Discussion

Lines 262-263: The significance in the sex ratio is a result. The authors should discuss what are the implications of the lack of significance.

Done

Lines 270-272: This is a great result that this study is the first to calculate “relative and absolute growth rates”. But what does that mean? What information do we get from the rates?

Absolute growth rates indicate actual growth between two years (ages) in terms of weight or length. Absolute growth rate decreases with age (t) and provide information about which years (ages) the growth is the highest. The way absolute growth rate is calculated it depends strongly on the size the fish has reached. So for comparison purposes, relative growth rates may be more useful. Relative growth rate is used to determine age-related growth rate in natural populations.